# Population-level deficit of homozygosity unveils *CPSF3* as an intellectual disability syndrome gene

Gudny A. Arnadottir [1,2], Asmundur Oddsson [1], Brynjar O. Jensson[1], Svanborg Gisladottir [3], Mariella T. Simon [4], Asgeir O. Arnthorsson[1], Hildigunnur Katrinardottir[1], Run Fridriksdottir[1], Erna V. Ivarsdottir [1], Adalbjorg Jonasdottir[1], Aslaug Jonasdottir[1], Rebekah Barrick[4], Jona Saemundsdottir[1], Louise le Roux [1], Gudjon R. Oskarsson[1], Jurate Asmundsson[5], Thora Steffensen[5], Kjartan R. Gudmundsson[1], Petur Ludvigsson[6], Jon J. Jonsson[2,3], Gisli Masson[1], Ingileif Jonsdottir [1,2], Hilma Holm [1], Jon G. Jonasson[2,5], Olafur Th. Magnusson[1], Olafur Thorarensen[6], Jose Abdenur[4], Gudmundur L. Norddahl[1], Daniel F. Gudbjartsson [1,7], Hans T. Bjornsson[2,3,8], Unnur Thorsteinsdottir[1,2], Patrick Sulem [1✉] & Kari Stefansson [1,2✉]

Predicting the pathogenicity of biallelic missense variants can be challenging. Here, we use a deficit of observed homozygous carriers of missense variants, versus an expected number in a set of 153,054 chip-genotyped Icelanders, to identify potentially pathogenic genotypes. We follow three missense variants with a complete deficit of homozygosity and find that their pathogenic effect in homozygous state ranges from severe childhood disease to early embryonic lethality. One of these variants is in *CPSF3*, a gene not previously linked to disease. From a set of clinically sequenced Icelanders, and by sequencing archival samples targeted through the Icelandic genealogy, we find four homozygous carriers. Additionally, we find two homozygous carriers of Mexican descent of another missense variant in *CPSF3*. All six homozygous carriers of missense variants in *CPSF3* show severe intellectual disability, seizures, microcephaly, and abnormal muscle tone. Here, we show how the absence of certain homozygous genotypes from a large population set can elucidate causes of previously unexplained recessive diseases and early miscarriage.

[1] deCODE Genetics/Amgen, Inc., Reykjavik, Iceland. [2] Faculty of Medicine, University of Iceland, Reykjavik, Iceland. [3] Department of Genetics and Molecular Medicine, The National University Hospital of Iceland, Reykjavik, Iceland. [4] Division of Metabolic Disorders, Children's Hospital of Orange County, Orange, CA, USA. [5] Department of Pathology, The National University Hospital of Iceland, Reykjavik, Iceland. [6] Children's Hospital Iceland, The National University Hospital of Iceland, Reykjavik, Iceland. [7] School of Engineering and Natural Sciences, University of Iceland, Reykjavik, Iceland. [8] Department of Genetic Medicine, Johns Hopkins University, Baltimore, MD, USA. ✉email: patrick.sulem@decode.is; kari.stefansson@decode.is

Genetically isolated populations that go through episodes of rapid growth are of interest to the study of the role of rare variants in the pathogenesis of the disease. In such populations, certain deleterious sequence variants can reach higher frequencies than in large outbred populations[1]. Whole-genome sequencing (WGS) of a large fraction of the Icelandic population has provided valuable insights into the genetic makeup of an isolated population, facilitating the detection of rare pathogenic genotypes[2,3]. This has motivated the first clinical sequencing effort in Iceland, mainly centered on rare disease cases referred by medical specialists at the National University Hospital of Iceland[2–6]. By now, we have performed clinical WGS for 764 patients and their family members. The current overall rate of detection of causative genotypes is 35%, comparable to other centers using next-generation sequencing[7], but around 460 patients remain without a clear genetic cause. Due to the isolation of the Icelandic population and the founder effect, some sequence variants that may be pathogenic in the homozygous states, reach elevated frequencies[2]. This holds the potential of uncovering previously unidentified causes of autosomal recessive (AR) diseases[8].

We have previously mapped out a large set of carriers of biallelic predicted loss-of-function (pLoF) variants in the Icelandic population, i.e., stop-gained, frameshift, or essential splice variants[9]. In doing so, we observed an overall deficit of double transmissions of pLoF variants from a pair of heterozygous parents to their offspring[9]. Such a deficit will appear if the homozygous genotype is expected to (A) be incompatible with early embryonic development (early miscarriage) (B) be incompatible with late embryonic development (late miscarriage, stillbirth), or (C) interfere with reproduction (e.g., by early death or severe disease). We recently characterized one of the genotypes from this set, a homozygous loss-of-function variant in CYBC1, as the cause of a previously unknown form of chronic granulomatous disease[6].

Most pLoF variants disrupt the protein and its function, and thus their pathophysiological roles are clear. For missense variants, however, only a subset disrupts the function of a protein in a way that leads to disease. Multiple methods have been developed to predict the pathogenicity of missense variants, taking into account evolutionary conservation of the sequence, physiochemical differences between amino acids, and expected effects on a protein's structure and biological function[10]. Moreover, population-genetic metrics based on the deficit of observed versus expected number of sequence variants, i.e., genic constraint scores, have shown great promise for highlighting genes where heterozygous missense variants are expected to be pathogenic[11–13]. However, predicting the pathogenicity of biallelic missense variants remains challenging[14].

Here we demonstrate that with a large dataset representative of an isolated population, it is possible to assess the pathogenicity of homozygous missense variants based on the deficit of homozygous carriers of such variants.

## Results

### The deficit of homozygous carriers outlines potentially pathogenic genotypes.
We searched for missense variants with a deficit of observed homozygous carriers, versus the expected number, in a population set of 153,054 Icelanders who were chip-genotyped and with rare variants imputed based on the WGS of 56,969 Icelanders. The population set mainly consists of adult volunteers, at a mean age of 58.5 years. We defined a complete deficit as missense variants present in the population set at a minor allele frequency (MAF) high enough to expect at least three homozygous carriers (MAF > 0.40%), assuming Hardy-Weinberg equilibrium, with no homozygous carriers observed. We found

114 missense variants meeting these criteria, 34 of which were in known AR Mendelian disease genes (Supplementary Data 1 and Supplementary Table 1). We observed that five of these variants presented in homozygous state in the clinical set (Table 1), a set consisting of 764 individuals previously WGS in a clinical context. Among these variants were c.1016 T > C (p.Leu339Pro) (RefSeq NM_001363118.2) in SLC52A2 (MAF = 0.60%), that has caused AR Brown-Vialetto van Laere syndrome type 2 in at least four genotypically confirmed homozygous carriers in Iceland[2], c.655 G > A (p.Gly219Arg) (RefSeq NM_177550.4) in SLC13A5 (MAF = 0.49%), that has caused epileptic encephalopathy in at least three homozygous carriers in Iceland, and c.557 A > C (p.Glu186Ala) (RefSeq NM_000404.3) in GLB1 (MAF = 0.52%), that has caused GM1-gangliosidosis in at least one homozygous carrier in Iceland.

### Homozygous missense variants in CPSF3 cause a severe intellectual disability syndrome.
The fourth variant, c.1403 G > A (p.Gly468Glu) (RefSeq NM_016207.3) in CPSF3, was found in homozygous state in two previously unresolved clinical cases (MAF = 0.41% in the population set; three expected homozygous carriers and none observed, Table 1). To our knowledge, sequence variants in CPSF3 had not yet been reported to cause a disease phenotype. The two homozygous carriers, whom we will refer to as patient A and patient B, were not closely related (meiotic distance = 12) but displayed strikingly similar phenotypes, see Table 2. Both patients were males, born at term to non-consanguinious parents. Both patients had an intellectual disability, with seizures from infancy, failure to thrive, microcephaly, nystagmus, and a severe delay in psychomotor development. Patient A died from pneumonia at 2 years of age, patient B is currently being treated for severe seizures (13 years old at last clinical evaluation). Analysis of the WGS data from both patients did not reveal any other plausible genetic causes (see Methods).

Based on the frequency of the p.Gly468Glu variant in the Icelandic population, we postulated that there could have been previous, unrecognized, homozygous carriers. The genealogical database constructed by deCODE genetics has near-complete genealogical information of Icelanders from the last century[2]. This allowed us to find three additional couples who were heterozygous carriers of p.Gly468Glu in the population set of 153,054 genotyped Icelanders, linked in the genealogical database by their offspring. For these couples, each offspring had a 25% probability of being a homozygous carrier of p.Gly468Glu. Combined, the three couples had ten offspring, four of whom died before eight years of age and had phenotypic features consistent with patients A and B, including intellectual disability (4/4), hypotonia/low muscle tone (4/4), seizures (3/4), and microcephaly (2/4) (Table 2). We were able to acquire paraffin-embedded archival samples from two of these children, patients C and D, siblings who had died at seven and four years of age (patient C 1960–1967; patient D 1964–1968). Whole-genome and Sanger sequencing of these samples confirmed a homozygous status of p.Gly468Glu in CPSF3 in both siblings. The siblings, whose parents were first cousins, were part of the same extended family as patient A (meiotic distance = 5, Supplementary Fig. 1). Importantly, analysis of WGS data from the siblings (see Methods), failed to identify any other pathogenic or likely pathogenic genotypes. Samples from the other two suspected homozygous Icelanders, patient E (1990–1995) and patient F (1955–1956), were not available (Table 2, Supplementary Fig. 2). However, the phenotypes of patients E and F, including intellectual disability, motor delay, and microcephaly, strongly suggests they have the same condition as the genotypically confirmed CPSF3 p.Gly468Glu homozygous patients.

**Table 1 Missense variants with a complete deficit of observed homozygous carriers, versus an expected number in the population set of 153,054 Icelanders, that present in homozygous state in the clinical WGS set.**

| Gene[a] | Position (hg38) | cDNA change | Protein change | MAF Iceland | MAF Europe[b] | O/E HMZ | AR disease listed on OMIM | ClinVar ID | Previously reported as HMZ |
|---|---|---|---|---|---|---|---|---|---|
| SLC2A2 | chr8:144360604 | c.1016 T > C | p.Leu339Pro | 0.60% | 0.01% | 0/6 | Brown-Vialetto-Van Laere syndrome, type 2 | 39577 | Yes |
| SLC13A5 | chr17:6703031 | c.655 G > A | p.Gly219Arg | 0.49% | 0.03% | 0/4 | Early infantile epileptic encephalopathy, type 25 | 140752 | Yes |
| GLB1 | chr3:33058265 | c.557 A > C | p.Glu186Ala | 0.52% | $2\times10^{-3}$% | 0/4 | GM1-gangliosidosis | NA | Yes |
| CPSF3 | chr2:9452920 | c.1403 G > A | p.Gly468Glu | 0.41% | NA | 0/3 | NA | NA | No |
| GNE | chr9:36227397 | c.1132 G > T | p.Asp378Tyr | 0.60% | 0.03% | 0/6 | GNE-myopathy | 283278 | No |

MAF minor allele frequency, O/E observed/expected, HMZ homozygous, AR autosomal recessive, NA not applicable.
[a]RefSeq transcripts are provided in Supplementary Data 1.
[b]Based on 64,603 WES/WGS non-Finnish European samples on gnomAD[13].

Subsequently, through GeneMatcher[15], we found another homozygous missense variant in *CPSF3*; c.1061 T > C (p.Ile354Thr) (Supplementary Table 2). The variant was discovered through the whole-exome sequencing (WES) of two patients, whom we will refer to as G and H, from the Children's Hospital of Orange County (CHOC). Patients G and H were first-degree cousins from a large consanguineous family of Mexican descent (Supplementary Table 3, Supplementary Fig. 3). Similar to what we observed in our Icelandic patient set, the cousins presented with intellectual disability, seizures, abnormalities of muscle tone, and microcephaly (Table 2). Furthermore, the analysis of their WES data did not yield any other pathogenic or likely pathogenic genotypes relating to their condition.

We do not observe any consistent phenotypic associations among 680 heterozygous carriers of *CPSF3* p.Gly468Glu in the Icelandic population set, nor do we observe a reduction in mRNA expression in white blood cells from 150 RNA sequenced heterozygous carriers (p-value = 0.91; Effect = 0.01 SD, Supplementary Fig. 4). Western blot analysis of EBV transformed lymphocytes from Patient B confirmed that the CPSF3 protein is expressed in a homozygous carrier (Supplementary Fig. 5). This indicates that p.Gly468Glu does not quantitatively affect the production of the CPSF3 protein, but would rather be expected to alter the function of the protein. The *CPSF3* gene is not constrained for pLoF variants (pLI=0.00) and not notably constrained for missense variants (z-score = 3.61), which is in line with what is expected for genes in which variants cause disease under an autosomal recessive mode of inheritance[16]. There is no homozygous carrier of a pLoF variant in *CPSF3* among 56,969 WGS Icelanders, nor among 141,456 WES or WGS individuals on gnomAD[13], although no pLoF variant reaches a high frequency in these datasets (MAFs below 0.02%).

In total, we have identified six patients who are homozygous for missense variants in *CPSF3*, as well as two suspected homozygous carriers, all of whom present with features of intellectual disability, seizures, microcephaly, and abnormal muscle tone.

**A homozygous missense variant in *GNE* causes a novel *GNE* allelic disorder.** The fifth missense variant presenting in homozygous state in the clinical set, with a complete deficit of homozygous carriers in the population set, was c.1132 G > T (p.Asp378Tyr) (RefSeq NM_005476.5) in *GNE* (MAF = 0.59%; six expected homozygous carriers and none observed, Table 1). The homozygous carrier was a female infant who had died shortly after birth, but had presented with acidosis, a ventricular septal defect, micro-Ebstein anomaly, and polysplenia. The presence of a myopathic disorder was suspected, and a biological sample was acquired postmortem. Prior to acquiring this biological sample, we performed WGS for her non-consanguineous parents (meiotic distance = 16), searching for potential recessive genetic causes for her condition. We observed that the parents were both heterozygous carriers of the *GNE* p.Asp378Tyr variant, which is a known pathogenic variant in *GNE*-myopathy[17,18]. *GNE*-myopathy, or Nonaka myopathy, is a progressive AR disease of muscle weakness with a typical age of onset between 20 and 40 years (ClinVar Variation ID 283278)[17,18]. Although initially we did not consider this genotype to align with the severe presentation of the female infant, WGS of the postmortem biological sample from her confirmed that she was indeed homozygous for *GNE* p.Asp378Tyr. Moreover, the parents were currently experiencing another pregnancy, and a 12-week ultrasound had shown increased nuchal thickness. WGS of a fetal sample from this pregnancy also showed homozygosity for p.Asp378Tyr. Importantly, WGS analysis of samples from both the deceased

**Table 2 Phenotypes observed among suspected and confirmed homozygous carriers of missense variants in *CPSF3*.**

| Individual ID | Variant[a] | Genotype | Age at last evaluation | Cause of death (age) | Neurological symptoms | Brain abnormalities | Craniofacial dysmorphism | Musculoskeletal symptoms | Immunological/GI symptoms | Other phenotypes |
|---|---|---|---|---|---|---|---|---|---|---|
| Patient A | p.Gly468Glu | HMZ | 2 years | Pneumonia (2) | Severe intellectual disability; Developmental delay; Seizures; Nystagmus | NA | Microcephaly; Short, upturned nose | Congenital hypotonia | Acute gastroenteropathy; Constipation; Repeated infections | Failure to thrive; Undescended testicle, bilateral |
| Patient B | p.Gly468Glu | HMZ | 13 years | NA | Intellectual disability; Developmental delay; Seizures; Cerebral palsy, dyskinetic; Nystagmus | Normal results from brain MRI and CT from first year of life | Microcephaly; Long face | Low muscle tone in whole body; Sparse calf muscles; Muscle cramps | Repeated bacterial and viral infections; GERD Constipation | Chronic nephritic syndrome; Under curve in height and weight; Flat foot |
| Patient C[b] | p.Gly468Glu | HMZ | 7 years | Pneumonia (7) | Intellectual disability; Seizures; Cerebral palsy; Strabismus | Cerebral atrophy (autopsy); Cerebral edema (autopsy) | Microcephaly; Long face | Hypotonia; Sparse muscles; Spastic paraplegia | Chronic otitis media; Constipation | Failure to thrive; Hypoplastic thyroid gland; Under curve in height; Undescended testicle (right side); Thought to be blind |
| Patient D[b] | p.Gly468Glu | HMZ | 4 years | Broncho-pneumonia (4) | Intellectual disability; Seizures | Cerebellar atrophy (autopsy); Degeneration of cerebral parenchyma (autopsy) | Microcephaly | Sparse muscles; Atonic paraplegia | NA | Under curve in height and weight; Thought to be blind; Hypoplastic ovaries (autopsy) |
| Patient E | p.Gly468Glu | Suspected HMZ[c] | 5 years | Respiratory failure (5) | Severe intellectual disability; Seizures; Extrapyramidal cerebral palsy; Strabismus | NA | Microcephaly; Dysmorphic features (unspecified) | Severe hypotonia; Motor delay; Spasticity | Bacteremia; Repeated invasive bacterial infections, otitis media, pulmonary infections | Feeding difficulties; Hypoplastic external genitalia |
| Patient F | p.Gly468Glu | Suspected HMZ[c] | 1 year | Pneumonia (1) | Intellectual disability; Spontaneous nystagmus | NA | Short, upturned nose; Macroglossia | Hypotonia; Motor delay | Repeated infections, otitis media; Suspected meningitis | Hypothyroidism (autopsy); Adenoma of adrenal gland (autopsy); Thought to be blind |
| Patient G | p.Ile354Thr | HMZ | 12 years | NA | Severe intellectual disability; Seizures (at 7 MOA); Peripheral motor neuropathy; Strabismus; Nystagmus | Cerebral atrophy; Thinning of corpus callosum; White-matter volume loss; Vermian hypoplasia; Hypoplastic cerebellum | Microcephaly; High palate; Protruding teeth | Spasticity (at 7 MOA); Motor delay; Inability to sit or crawl; Hip contractures; Bilateral hip dislocation | GERD | Feeding difficulties; Optic atrophy; Cortical blindness; Decreased fetal movements |
| Patient H | p.Ile354Thr | HMZ | 13 years | NA | Severe intellectual disability; Seizures, generalized tonic-clonic seizures; Peripheral motor neuropathy; Strabismus; Nystagmus | White-matter volume loss; Dilatation of lateral ventricles; Atrophic corpus callosum; Small cerebellar hemispheres | Microcephaly; High palate; Protruding teeth | Motor delay; Inability to sit or crawl | GERD | Feeding difficulties; Bilateral pallor of optic nerves; Cortical blindness; Decreased fetal movements |

*GI* Gastrointestinal symptoms, *HMZ* homozygous, *NA* not applicable/not assessed, *GERD* gastroesophageal reflux disease, *MOA* months of age.
[a]RefSeq transcript NM_016207.3.
[b]Patients C and D were siblings.
[c]Parents both heterozygous carriers of *CPSF3* p.Gly468Glu, samples from patients unavailable for confirmation of homozygous genotype.

female infant and the new fetal sample failed to produce any other pathogenic or likely pathogenic genotypes.

Over 180 pathogenic variants in *GNE* have been described since the genetic basis for *GNE*-myopathy was first established in 2001[18,19]. The majority of reported pathogenic genotypes in *GNE* consist of two missense variants, and to date, no patient with a biallelic pLoF genotype (consisting of stop-gained, frameshift, or essential splice variants) has been reported[17]. This is in line with observations from mouse studies; where complete knockouts of *GNE* result in embryonic lethality[20]. Although *GNE* p.Asp378Tyr is a known pathogenic variant, it has never before been reported in the homozygous state, only as part of compound heterozygous genotypes with other variants in *GNE*[17,18]. Interestingly, patients with a compound heterozygous genotype including p.Asp378Tyr, have been described to have more severe and earlier onset of symptoms than other *GNE*-myopathy patients[19,21]. Moreover, experimental studies have shown that p.Asp378Tyr has a more drastic effect on GNE enzymatic activity than most other known pathogenic *GNE* variants; a 60% reduction of epimerase activity and 50% reduced kinase activity[22].

We found that among nine couples where both individuals are heterozygous carriers of p.Asp378Tyr, six women have a medical history of miscarriage (66.7%; Table 3). This is significantly higher than the 26.7% of 74,731 Icelandic women with a history of miscarriage ($P = 0.0090$, OR = 6.0; logistic regression adjusting for mother's year of birth). Furthermore, the observed rate of miscarriage, calculated as the ratio between miscarriages and pregnancies for each mother, is higher among the *GNE* p.Asp378Tyr carrier couples (mean ratio is 0.29 for carriers compared to 0.07 in the population, $P = 8.3 \times 10^{-7}$; linear regression adjusting for mother's year of birth).

**A homozygous missense variant in *GLE1* causes early miscarriage.** Encouraged by the apparent pathogenicity of the two missense variants, *CPSF3* p.Gly468Glu and *GNE* p.Asp378Tyr, in the homozygous state, we looked into other missense variants showing a complete deficit of homozygous carriers in the population set (Supplementary Data 1). Among these variants was c.1706G > A (p.Arg569His) (RefSeq NM_001003722.2) in *GLE1* (MAF = 0.82%), with ten homozygous carriers expected in the population set, but none observed (Table 3). *GLE1* p.Arg569His has been described as pathogenic when part of compound heterozygous genotypes with other variants in *GLE1*, causing AR lethal congenital contracture syndrome 1 (LCCS1, MIM 253310)[8,23,24]. This severe condition, as well as the other *GLE1*-related disease; congenital arthrogryposis with anterior horn cell disease (CAAHD, MIM 611890), leads to death in the perinatal period in the majority of identified cases[23]. Similar to *GNE* p.Asp378Tyr, *GLE1* p.Arg569His has never been reported in the homozygous state.

In our population set, we identified 17 couples where both individuals were heterozygous carriers of p.Arg569His. Despite an evident deficit of homozygous carriers of p.Arg569His, we did not observe premature death or severe disease among the offspring of these 17 carrier couples. Moreover, we did not detect a significant association of p.Arg569His with clinically confirmed miscarriage ($N = 19,965$ mothers with a history of miscarriage and $N = 54,766$ controls, $P = 0.02$; OR = 1.2). Considering the high deficit of homozygous carriers of *GLE1* p.Arg569His, we postulated that homozygosity for the variant would result in lethality in the first trimester (before week 13), that would be under-registered in medical records. We investigated medical records specific to early miscarriage, including subsequent childbirth records and interviews, from 11 women out of the 17 carrier couples. Seven (63.6%) mentioned one or more

spontaneous abortions at an approximate gestational age of 5–8 weeks. In comparison, reported rates of miscarriage are 12–24% for all clinically confirmed pregnancies[25,26]. The high incidence of early miscarriage among *GLE1* p.Arg569His carrier couples, along with the deficit of homozygous carriers, strongly suggests that homozygosity for p.Arg569His causes early embryonic death.

## Discussion

We describe a set of missense variants with a complete deficit of observed homozygous carriers in a set representative of the adult Icelandic population. Guided by the homozygote deficit, we discovered that missense variants in *CPSF3* cause a severe intellectual disability syndrome in homozygous carriers. Prior to this current work, sequence variants in *CPSF3* were not reported to cause disease. We describe six *CPSF3* patients in Iceland and the US, who are homozygous for one of two missense variants; p.Gly468Glu and p.Ile354Thr. Near-complete genealogical information for the Icelandic population set allowed us to discover previous, unrecognized, homozygous carriers of the Icelandic variant, p.Gly468Glu. Based on the genealogical information we found three p.Gly468Glu carrier couples who, combined, had four children who had died prematurely. We managed to test archival samples for two of these children, and confirm a homozygous genotype.

*CPSF3* encodes Cleavage And Polyadenylation Specificity Factor 3 (CPSF3 or CPSF73), a 73 kDa subunit of the Cleavage and Polyadenylation Specificity Factor (CPSF) complex[27]. Cleavage and polyadenylation of the 3′ end of precursor mRNAs (pre-mRNAs) are necessary before transport out of the nucleus. The CPSF complex plays a crucial role in this 3′ end cleavage, recognizing the AAUAAA cleavage site and interacting with other factors to implement cleavage[27]. CPSF3 has endonuclease activity and is predicted to catalyze cleavage just downstream of the AAUAAA site[27]. It is also thought to act as both the endonuclease and 5′−3′ exonuclease in histone pre-mRNA processing[28]. The CPSF3 protein consists of 684 amino acids, with two metallo-β-lactamase domains and a conserved β-CASP domain, thus belonging to the metallo-β-lactamase family of zinc-dependent hydrolases[29] (Fig. 1). The p.Gly468Glu variant causes substitution of the small and nonpolar amino acid glycine with glutamic acid, a large negatively charged residue, a change that is likely to affect the structure of the CPSF3 protein. The p.Ile354Thr variant occurs within the β-CASP domain of *CPSF3*, a domain that controls access to the active site of CPSF3, whereas p.Gly468Glu occurs in a linker domain between the highly conserved second metallo-β-lactamase domain of CPSF3 and the C-terminal domain (Fig. 1). *Cpsf3* knockout mice show early embryonic lethality, and *Cpsf3* is homologous to the human *CPSF3* with 98.5% amino acid identity[30]. Yeast studies have shown that mutations of key residues in the CPSF3 yeast homolog (53% amino acid identity), including within the β-CASP domain, are lethal[27]. This suggests that a drastic phenotypic effect will be observed not only from a complete loss of the CPSF3 protein, as seen in knockout mice but also from biallelic variants affecting its enzymatic function. Both p.Gly468Glu and p.Ile354Thr occur at genomic positions highly conserved across mammalian species, with GERP scores of 5.81 and 5.78, respectively (corresponding to the top 96th percentile of all coding variants)[31].

*CPSF3* is the first member of the cleavage and polyadenylation specificity factor complex to be associated with a severe Mendelian disorder. However, heterozygous variants in the gene *CPSF1*, encoding cleavage and polyadenylation specificity factor 1, have recently been linked to autosomal dominant early-onset high myopia[32]. A 2019 study presented an inhibitor of CPSF3,

**Table 3 Two missense variants that have a complete deficit of homozygous carriers in the population set of 153,054 Icelanders, and are likely causing miscarriage in the homozygous state.**

| Gene[a] | Variant | MAF Iceland | MAF Europe[b] | O/E HMZ | ClinVar ID | Previously reported as HMZ | N carrier couples[c] | N miscarriage > week 12 | N miscarriage < week 12 |
|---|---|---|---|---|---|---|---|---|---|
| GNE | p.Asp378Tyr | 0.60% | 0.03% | 0/6 | 283278 | No | 9 | 6/9 (66.7%) | 1/5 (20.0%) |
| GLE1 | p.Arg569His | 0.82% | 0.05% | 0/10 | 6463 | No | 17 | 0 | 7/11 (63.6%) |

MAF minor allele frequency, O/E observed/expected, HMZ homozygous.
aRefSeq transcripts are provided in Supplementary Data 1.
bBased on 64,603 WES/WGS non-Finnish European samples on gnomAD[13].
cCarrier couples were identified in the set of 153,054 chip-genotyped Icelanders.

JTE-607, a small molecule that was developed as an inflammatory cytokine inhibitor but had an unknown mode of action until its interaction with CPSF3 was discovered[33]. Inhibition of CPSF3, and blocking the release of newly synthesized pre-mRNAs, can enforce cellular apoptosis and has been considered as potential cancer treatment[34,35]. Importantly, however, modulation of mRNA processing in cancer cell lines is vastly different from a genetic loss of mRNA processing. Nonetheless, these studies of CPSF3 inactivation underline the critical role of the protein in the mRNA processing pathway.

In isolation, CPSF3 has weak enzymatic activity, and interaction with other factors of the cleavage and polyadenylation machinery is necessary for efficient cleavage[29]. We hypothesize that the two missense variants we describe, p.Gly468Glu and p.Ile354Thr, affect the ability of CPSF3 to bind to other factors involved in the pre-mRNA cleavage mechanism. This effect could disrupt the pre-mRNA cleavage mechanism, resulting in the accumulation of unprocessed pre-mRNA in cells, which would in turn activate a cellular stress response[36]. Taken together, the crucial biological function of CPSF3 in humans, the prior evidence from animal models, and the consistency of the phenotype among the six confirmed homozygous patients, all support the role of the two CPSF3 variants, p.Gly468Glu and p.Ile354Thr, in the intellectual disability syndrome observed among homozygous carrier patients. Ultimately, follow-up functional studies are needed to document the effects of p.Gly468Glu and p.Ile354Thr on CPSF3 function.

In addition to CPSF3, we also discovered that a missense variant in GNE, p.Asp378Tyr, causes an early-onset myopathic disease in homozygous carriers. The variant has been reported as part of compound heterozygous genotypes in patients with an adult-onset myopathic disorder known as GNE-myopathy. In contrast, we find that in a homozygous state it causes a drastic disease with cardiac complications, resulting in death in the perinatal period. We also observe an elevated rate of miscarriage among p.Asp378Tyr carrier couples. GNE encodes glucosamine (UDP-N-acetyl)-2-epimerase/N-acetylmannosamine kinase, an enzyme that catalyzes sialic acid biosynthesis[37]. Animal studies have suggested that a reduction in sialic acid production is tolerated to a certain threshold, whereas total knockout of GNE and subsequent complete loss of endogenous sialic acid production is embryonic lethal[37]. Based on a complete deficit of homozygous carriers of p.Asp378Tyr in the Icelandic population set, an increase in miscarriage among p.Asp378Tyr carrier couples, and confirmation of the variant in homozygosity in two symptomatic cases, we suggest that homozygosity for p.Asp378Tyr causes a reduction in sialic acid production that is below the critical sialylation threshold necessary for early human development. To our knowledge, this is the first report of a pathogenic GNE genotype resulting in such a severe phenotype.

Finally, we describe how homozygosity for p.Arg569His in GLE1 leads to early embryonic lethality, manifesting as first-trimester miscarriage for mothers from carrier couples, that has until now remained unidentified. GLE1 p.Arg569His has previously only been observed as part of a compound heterozygous genotype with other variants in GLE1, causing a severe neonatal condition. GLE1 encodes GLE1 RNA export mediator, a protein required for the transport of polyadenylated mRNA from the nucleus to the cytoplasm[38]. Efficient processing and transport of nuclear mRNA is essential for proper eukaryotic gene expression, and has been implicated in several human diseases[39]. Two of the genes we describe, GLE1 and CPSF3, encode proteins with a crucial role in the export of nuclear mRNA, underlining the importance of this biological process in early human development.

In conclusion, we used a deficit of observed homozygous carriers versus their expected number in a large population set to identify pathogenic missense genotypes. We believe our approach may be

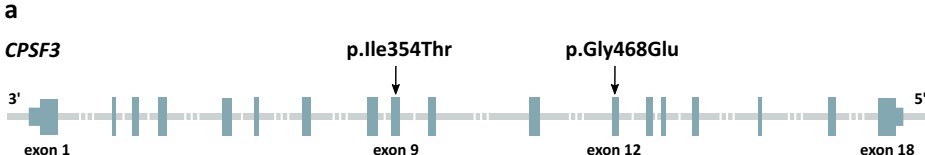

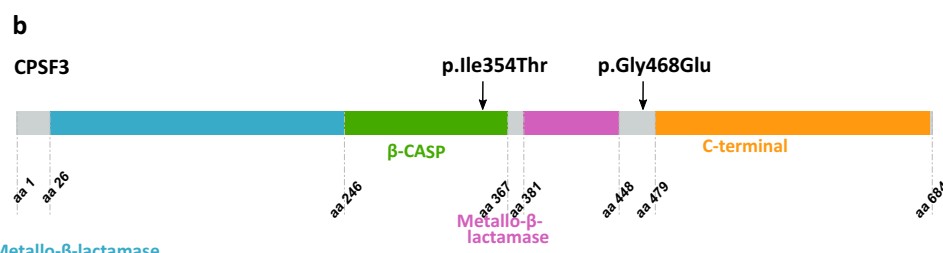

**Fig. 1 Structure of** *CPSF3* **(RefSeq transcript NM_016207.3) and location of the two missense variants found in homozygous state in patients with a severe intellectual disability syndrome. a** The p.Ile354Thr and p.Gly468Glu variants are located in exon 9 and 12 of *CPSF3*, respectively. Dark gray sections represent exons and untranslated regions, light gray lines represent introns. **b** The p.Ile354Thr variant is in the β-CASP domain of CPSF3 (amino acids 246–367, shown in green). The p.Gly468Glu variant is located within an uncharacterized linker domain (amino acids 448–479) between the critical second metallo-β-lactamase domain (shown in pink) and the highly conserved C-terminal domain (amino acids 479–682, shown in yellow) of CPSF3. The first metallo-β-lactamase domain is shown in blue, uncharacterized domains are shown in gray.

helpful in other large population-based genetic datasets, especially in light of recent data suggesting as high as 1% of unrelated European couples are at risk of having offspring with a severe AR genetic disease[40]. Until now, the study of genotype-phenotype correlation has mainly been centered on observed genotypes. We show that the absence of homozygous genotypes from a large population set, given an expected number, can elucidate the causes of previously unexplained AR diseases and early miscarriage.

## Methods

**Population set**. Our approach to WGS, genotyping, long-range phasing, and imputation of a substantial fraction of the Icelandic population has been described in detail in previous publications[2,41]. In brief, 56,959 Icelanders participating in various disease projects at deCODE genetics have been WGS using standard TrueSeq methodology (Illumina), to a median depth of 37X, and genotyped with Illumina microarrays (chip-genotyped). An additional 96,095 Icelanders have been chip-genotyped and not WGS. Genotypes of sequence variants identified through sequencing (SNPs and indels) have been imputed into all chip-typed Icelanders, resulting in a set of 153,054 chip-genotyped and imputed Icelanders. The set consists mainly of adult individuals, 88% were alive at the time of this study, the mean age of living individuals was 58.5 years, and 54% were female. For the purpose of this study, individuals with either one or both parents of foreign ancestry were removed from the set.

All participating individuals who donated blood or buccal tissue samples, or their guardians, provided written informed consent. All sample identifiers were encrypted in accordance with the regulations of the Icelandic Data Protection Authority. Personal identities of the participants and biological samples were encrypted by a third-party system approved and monitored by the Icelandic Data Protection Authority. The study was approved by the Data Protection Authority (ref. 2013030423/ÞS/−, with amendments) and the National Bioethics Committee (ref. VSN-19-023, VSNb2019010015/03.01), which also reviewed and approved the protocol, methodology, and all documents presented to the participants. All methods were performed in accordance with the relevant guidelines and regulations.

**Clinical set**. For the purpose of clinical WGS we have recruited 764 patients and their family members, amounting to a total of 2501 individuals. These individuals were recruited in the period 2014–2021, the mean age of the patients was 26.4 years in 2021, and 46% were female. Samples from patients and family members, submitted to WGS, were from whole-blood, buccal tissue, or purified DNA. WGS and genotyping of the set of 2501 were performed as described above. For the purpose of this study, all individuals recruited on the basis of clinical WGS were removed from the Icelandic population set.

**Defining homozygous missense variants of interest**. Based on a given MAF of a sequence variant, a homozygous carrier frequency can be calculated assuming the Hardy-Weinberg equilibrium. We took variants with a MAF corresponding to an expected homozygous carrier count of three in the population set of 153,054 chip-genotyped and imputed Icelanders. This corresponded to sequence variants with a MAF over 0.40%. We considered only genotypes with imputation information over 0.9, excluding variants on the X chromosome.

**Analysis of WGS data for clinical purposes**. Sequence variants were called using the Genome Analysis Toolkit (GATK)[42], reads were called with GATK using a multi-sample configuration. Variants were annotated using release 100 of the Variant Effect Predictor (VEP-Ensembl), using RefSeq gene annotations[43]. To be able to filter out genotypes over a certain frequency threshold we used allelic frequencies from phased genotypes of 30 million SNPs and INDELs from 56,959 WGS Icelanders[2,41]. Additional frequency filtering was performed using alleles from publicly available datasets of the genome Aggregation Database[13]. For the purpose of identifying potentially pathogenic sequence variants we looked at rare (MAF < 2%) coding and splicing variants, including variants with predicted high (stop, frameshift, and splice essential) and moderate (missense, inframe, and splice region) impact on protein function. We focused on single-nucleotide polymorphisms (SNPs) and small indels (<20 base pairs). We prioritized the analysis of the WGS data on variants in phenotypically relevant genes, i.e., genes with a known link to Mendelian disorders (according to Online Mendelian Inheritance in Men) that fit the phenotypes of interest. All candidate sequence variants were submitted to the ACMG criteria for classification as likely benign, benign, variants of uncertain significance (VUS), likely pathogenic, or pathogenic[44].

**Validation by Sanger sequencing**. Samples from all individuals homozygous for p.Gly468Glu in *CPSF3* and p.Asp378Tyr in *GNE*, along with their first-degree family members, were Sanger sequenced for confirmation of their respective genotypes. Primers for Sanger sequencing were designed using the Primer 3 software[45]. Following PCR, cycle sequencing reactions were performed in both directions on MJ Research PTC-225 thermal cyclers, using the BigDye Terminator Cycle Sequencing Kit v3.1 (Thermo Fisher Scientific, Waltham, MA, USA) and Ampure XP and CleanSeq kits (Beckman Coulter, High Wycombe, UK) for cleanup of the PCR products and cycle sequencing reactions. Sequencing products were run on the 3730 XL DNA Analyzer (Applied Biosystems™, Thermo Fisher Scientific) and analyzed with the Sequencher 5.4 software (Gene Codes Corporation, Ann Arbor, MI, USA).

**Protein analysis**. PBMCs were isolated from a venous blood sample from one *CPSF3* p.Gly468Glu homozygous carrier (patient B) via standard Ficoll-Paque (GE Health, Uppsala, Sweden, #17144002) density gradient centrifugation at 800 × *g* for 15 min in 50 ml Blood-Sep spin tubes (DACOS, Denmark, #037100SI) and cryopreserved in liquid nitrogen. Thawed cells were transformed using Epstein Barr virus. For western blot analysis, EBV transformed cells were lysed in RIPA buffer

(150 mM sodium chloride, 1% Triton X-100, 0.5% sodium deoxycholate, 0.1% SDS, 50 mM Tris pH 8), run on 4–12% gel (Invitrogen, Kiryat Shmona, Israel, #NW04127BOX) and blotted on a PVDF membrane (Novex, #IB24002). The membrane was blocked, then stained with CPSF3 (Abcam, #ab72295) and beta actin (Abcam, #ab6276) primary antibodies in 5% milk in TBS-T (200 mM Tris, 1500 mM NaCl, 0.1% Tween-20) overnight and mouse (LI-COR, 926–32212) and rabbit (LI-COR, #926–68073) secondary antibodies in TBS-T for 1 h. The membrane was scanned using the Odyssey infrared imaging system (LI-COR Biosciences).

**Reporting summary**. Further information on research design is available in the Nature Research Reporting Summary linked to this article.

## Data availability
Our previously described Icelandic population whole-genome sequence data[3] have been deposited at the European Variation Archive under accession PRJEB15197. All other data supporting the findings of this study are available within the article and its supplementary files.

## Code availability
We used the following publicly available software for analysis of the WGS data: BWA-MEM version 0.7.10, available at https://github.com/lh3/bwa, GenomeAnalysisTKLite version 2.3.9, available at https://github.com/broadgsa/gatk/, Picard tools version 1.117, available at https://broadinstitute.github.io/picard/, SAMtools version 1.3, available at http://samtools.github.io/, and Variant Effect Predictor (release 100), available at https://github.com/Ensembl/ensembl-vep. For association testing we used BOLT-LMM version 2.1, available at https://data.broadinstitute.org/alkesgroup/BOLT-LMM/downloads/. We used R version 3.6.0 to analyze data and create plots.

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

## Acknowledgements
We wish to thank the patients and family members followed in this study, and all other individuals who participated in the study and whose contribution made this work possible. M.T.S. was supported by a grant from the Sacchi Foundation.

## Author contributions

G.A.A., A.O., H.T.B., U.T., P.S. and K.S. designed the study. G.A.A., B.O.J., H.K., R.F. and P.S. carried out or supervised analysis of clinical WGS data. S.G. carried out analysis of medical records. S.G., M.T.S., R.B., P.L., J.J.J., O.Th., Jo.A. and H.T.B. met with patients and families involved in the study. A.O.A. and G.L.N. carried out or supervised laboratory experiments. J.S., L.R., Ju.A.,Th.S., I.J., H.H. and J.G.J. carried out or supervised the collection of biological samples. G.M. and D.F.G. developed the methods for data analysis. O.Th.M., Ad.J. and As.J. carried out sequencing. G.A.A., A.O., E.V.I., G.R.O., K.R.G., D.F.G. and P.S. performed statistical and bioinformatics analyses. G.A.A., A.O., B.O.J., D.F.G., H.T.B., U.T., P.S. and K.S. drafted the manuscript. All authors contributed to the final version of the paper.

## Competing interests

Authors affiliated with deCODE genetics/Amgen declare competing interests as employees. The remaining authors declare no competing interests.
