## [Peer Review File · Nature Communications]

Population-level deficit of homozygosity unveils CPSF3 as an intellectual disability syndrome geneReviewers' Comments:

Reviewer #1:

Remarks to the Author:

Using WGS and microarrays techniques, Arnadottir and colleagues report three missense variants (c.1403G>A, p.Gly468Glu in exon 12 of CPSF3 gene in linker domain, and c.1706G>A, p.Arg569His in GLE1 gene, and c.1225G>T, p.Asp409Tyr in GNE gene), that demonstrate a complete deficit of homozygous carriers in a set of 153,054 chip-genotyped and imputed Icelanders, and find that their pathogenic effect in homozygous state ranges from severe childhood disease to early embryonic lethality. Also they found homozygous missense variant c.1706G>A (p.Arg569His) in GLE1 causes early miscarriage. The crucial biological function of CPSF3 in humans, the prior evidence from animal models, and the consistency of the phenotype among the six confirmed homozygous patients in present deCODE study, all support CPSF3 as a novel intellectual disability syndrome gene.

The article has a sufficient impact and adds to the knowledge base. The research is novel, and the state of the art well described and the knowledge gap clearly defined. The applied research methodology is solid; Research is sound and any methodology described in the text is sustainable and reproducible. The results are reliable and the objectives have been reached and the conclusions are supported by the data and justified. However, some minor correction need to be done by authors listed below:

- Manufacturer, City, County or State, Country for buffers, kits should be added.
- All of the nucleotide changes should be followed by protein changes first mentioned in the text.
- Please note that acronyms must be written out in full in the Abstract and the Text the first time they are mentioned and used thereafter except at the beginning of a sentence.
- References need corrections; title of article (with first word only starting in capitals), Some references without page range, for example; references 18, 24, and 29.

Reviewer #2:

Remarks to the Author:

Arnadottir et al. identify likely causal homozygous missense variants in five genes, via the examination of WGS data from patients, without a genetic diagnosis, and by leveraging population-based data to prioritise variants, with an apparent deficit of homozygous carriers. One of the identified variants, occurred in two unrelated individuals, with a similar phenotype; identification of further cases (and likely cases), provided additional evidence that this was a novel ID gene.

I congratulate the authors on a very nice paper. This is a neat study design, made possible by the extensive and unique deCODE resource. The identification of CPSF3 is compelling. I have no major questions regarding the manuscript, just a couple of general queries / suggestions to the authors:

The 119 genes identified, where at least one missense variant had a deficit of homozygous carriers - are these genes generally constrained for missense variants (and/or pLoF variants), compared to other genes? There is some discussion of the lack of pLoF variants in CPSF3; have the authors looked at this in a systematic way, for all 119 genes?

There are 3 genes (Supp Table 1), identified as harbouring more than one missense variant with a deficit of observed homozygous carriers, and thus appear particularly intolerant to homozygous missense variation. For these genes, have the authors looked for potential compound heterozygotes amongst the clinical set? This could be extended to compound hets in all 119 genes comprising: 1. the variant deficient of hom carriers; and 2. any other rare missense variant in the gene. Examples of this scenario are stated in the manuscript, for the GNE and GLE1 variants, providing a precedent for this.

Why was the GNE variant not identified through standard analyses of the WGS data, given this is a rare (ie MAF <2%) homozygous variant, in an established disease gene? Whilst not an exact

phenotypic match, I'd expect to have been identified as a candidate?

The CPSF3 variant appears to be on a haplotype that is unique to the Icelandic population. Have the authors considered estimating the age of this variant?

Author's response to reviewers' comments

Reviewer #1 (Remarks to the Author):

Using WGS and microarrays techniques, Arnadottir and colleagues report three missense variants (c.1403G>A, p.Gly468Glu in exon 12 of CPSF3 gene in linker domain, and c.1706G>A, p.Arg569His in GLE1 gene, and c.1225G>T, p.Asp409Tyr in GNE gene), that demonstrate a complete deficit of homozygous carriers in a set of 153,054 chip-genotyped and imputed Icelanders, and find that their pathogenic effect in homozygous state ranges from severe childhood disease to early embryonic lethality. Also they found homozygous missense variant c.1706G>A (p.Arg569His) in GLE1 causes early miscarriage. The crucial biological function of CPSF3 in humans, the prior evidence from animal models, and the consistency of the phenotype among the six confirmed homozygous patients in present deCODE study, all support CPSF3 as a novel intellectual disability syndrome gene.

The article has a sufficient impact and adds to the knowledge base. The research is novel, and the state of the art well described and the knowledge gap clearly defined. The applied research methodology is solid; Research is sound and any methodology described in the text is sustainable and reproducible. The results are reliable and the objectives have been reached and the conclusions are supported by the data and justified. However, some minor correction need to be done by authors listed below.

Answer: We thank the reviewer for the positive comments on our manuscript, and for pointing out corrections to be made. We address each point directly below.

- **Manufacturer, City, County or State, Country for buffers, kits should be added.**

Answer: We have added manufacturer information to the chapters on Sanger sequencing and Protein analysis in the Methods section (page 17).

- **All of the nucleotide changes should be followed by protein changes first mentioned in the text.**

Answer: We have made changes throughout the manuscript according to the reviewer's suggestion, adhering to recommendations from the HGVS so that for first mentions of any of the variants discussed in our manuscript we present the nucleotide change, followed by the protein change, and provide the RefSeq transcript in parentheses. Specifically, we made changes for the first mentions of c.1016T>C (p.Leu339Pro) in SLC52A2 (line 89, page 5), for c.655G>A (p.Gly219Arg) in SLC13A5 (line 92, page 5), for c.557A>C (p.Glu186Ala) in GLB1 (line 93, page 5), for c.1403G>A (p.Gly468Glu) in CPSF3 (line 97, page 5), for c.1061T>C (p.Ile354Thr) in CPSF3 (line 131, page 6), and for c.1706G>A (p.Arg569His) in GLE1 (line 200, page 9). In reviewing our use of HGVS suggested nomenclature, we decided to switch to the newest set of RefSeq transcripts. This led to changes in our variant set, and

Supplementary Table 1 has been updated accordingly. This also led to the decision of using the transcript NM_005476.5 for GNE, so that we now refer to the GNE variant as c.1132G>T (p.Asp378Tyr) (changed in line 157, page 7, and throughout the manuscript when referring to the variant). After all first mentions of the variants we use protein changes to refer to them, and made sure to be consistent throughout the manuscript. We have also removed mentions of transcripts from Table 1, now pointing to Supplementary Table 1, and have moved all other mentions of transcripts in tables and figures to the footnotes (Figure 1, Table 2, Table 3, and Supplementary Table 3).

• Please note that acronyms must be written out in full in the Abstract and the Text the first time they are mentioned and used thereafter except at the beginning of a sentence.

Answer: We have made sure the correct use of acronyms throughout our text. Specifically, we corrected our use of the acronym WGS for whole-genome sequencing (changes made in the abstract, and in line 43, page 3), in writing out loss-of-function (line 61, page 3) when not referring to predicted loss-of-function which we defined as the acronym pLoF, and when defining the acronym MAF for minor allele frequency (line 84, page 4), which we then use throughout the text.

• References need corrections; title of article (with first word only starting in capitals), Some references without page range, for example; references 18, 24, and 29.

Answer: We have corrected the references with respect to these concerns.

Reviewer #2 (Remarks to the Author):

Arnadottir et al. identify likely causal homozygous missense variants in five genes, via the examination of WGS data from patients, without a genetic diagnosis, and by leveraging population-based data to prioritise variants, with an apparent deficit of homozygous carriers. One of the identified variants, occurred in two unrelated individuals, with a similar phenotype; identification of further cases (and likely cases), provided additional evidence that this was a novel ID gene.

I congratulate the authors on a very nice paper. This is a neat study design, made possible by the extensive and unique deCODE resource. The identification of CPSF3 is compelling. I have no major questions regarding the manuscript, just a couple of general queries / suggestions to the authors:

Answer: We thank the reviewer for positive feedback on the manuscript. We address each query/suggestion below.

The 119 genes identified, where at least one missense variant had a deficit of homozygous carriers - are these genes generally constrained for missense variants (and/or pLoF variants), compared to other genes? There is some discussion of the lack of pLoF variants in CPSF3; have the authors looked at this in a systematic way, for all 119 genes?

Answer: What is commonly referred to as genetic constraint is assessed by comparing the number of „expected“ heterozygous variants versus the number of „observed“ heterozygous variants over a given region (e.g. gene). This is what the gnomAD group has attempted to portray with their constraint scores for missense variants (z-scores) and predicted loss-of-function variants (pLI scores). Inherently, because these scores are obtained from heterozygous variant counts, they are poor predictors of autosomal recessive disease genes. This is reviewed nicely in a recent publication by Dawes et al. (PMID: 30993004). For the 119 genes we discuss, we see that on average they have a missense z-score of 0.43 (± 1.63) and a pLI score of 0.20 (± 0.37), so overall, they are not more constrained for coding variation than other genes (the average z-score for all 20K gnomAD genes is 0.76 ± 1.34 and pLI is 0.24 ± 0.38).

The reviewer points out that we discuss the lack of pLoF variants in CPSF3 in our manuscript. When we discuss the number of pLoF's in CPSF3 we are not implying that they are fewer than expected, but rather wanted the reader to get an idea of the coding diversity over CPSF3. To avoid misunderstanding, we have now made slight modifications of this text (page 7):

„The CPSF3 gene is not constrained for pLoF variants (pLI=0.00) and not notably constrained for missense variants (z-score=3.61), which is in line with what is expected for genes in which variants cause disease under an autosomal recessive mode of inheritance¹⁶. There is no homozygous carrier of a pLoF variant in CPSF3 among 56,969 WGS Icelanders, nor among 141,456 WES or WGS individuals on gnomAD¹³, although no pLoF variant reaches a high frequency in these datasets (MAFs below 0.02%).“

There are 3 genes (Supp Table 1), identified as harbouring more than one missense variant with a deficit of observed homozygous carriers, and thus appear particularly intolerant to homozygous missense variation. For these genes, have the authors looked for potential compound heterozygotes amongst the clinical set? This could be extended to compound hets in all 119 genes comprising: 1. the variant deficient of hom carriers; and 2. any other rare missense variant in the gene. Examples of this scenario are stated in the manuscript, for the GNE and GLE1 variants, providing a precedent for this.

Answer: We thank the reviewer for pointing this out. After having looked into this, we find that in one of the genes (i.e. HCN2) harboring two missense variants with a complete deficit of homozygous carriers, the two variants are in fact part of the same multinucleotide polymorphism since the two SNPs were representing consecutive nucleotides. We have now merged the two variants into one, and amended the corresponding amino acid change (Supplementary Table 1). For the two other genes harboring more than one missense variants (i.e. PCDH7 and PCDH8), this revision gave us an opportunity to discover a high homology between this group of genes (known as protocadherins). After revision, we decided to remove all protocadherin genes from our study, including PCDH7, PCDH8, PCDH11, and PCDH15, due to the high homology between them and uncertainty of the genotypes called in these homologous regions. We have now updated the variant count in the main text, now at 115 variants in an equal number of genes (line 85, page 4).

The reviewer also raises a fair question regarding assessing potential compound heterozygotes in the clinical set. We would like to provide a three-fold answer to this question:

A) *As part of the standard approach to clinical WGS analysis we have, naturally, assessed all possible compound heterozygous genotypes in known AR disease genes that are relevant to the case in question.*

B) *Performing an overall search for compound heterozygous genotypes among the 119 genes, that would consist of one of the deficit variants and other rare missense variants, is to our belief beyond the scope of this study. What the deficit of homozygous carriers allows us to do is outline potentially pathogenic missense variants, since the absence of these variants in homozygous state indicates they have some detrimental effect. We are not claiming that all variants presented in Supplementary Table 1 (among the 119 genes) are pathogenic, to do so requires an additional level of evidence, that we have provided in selected examples (CPSF3, GNE, and GLE1). Equating the effect of other rare missense variants in the same genes with the effect of these deficit genotypes would, to our belief, take away from the overall aim of our study.*

In addition, assessing pathogenicity of individual missense variants when they are part of compound heterozygous genotypes is a known problem. For compound heterozygous genotypes consisting of two distinct missense variants, the combination of both variants is what causes an effect at the protein level, and the exact role of each variant is unclear. In contrast, homozygous missense genotypes have the benefit of having only one variant contributing to the effect, and so the role of that single variant is easily understood.

C) For CPSF3, we have looked for instances where p.Gly468Glu is coupled with another rare missense variant in an individual from the clinical set, but find no such instances. We note that combined, other missense variants in CPSF3 have a cumulative MAF of 0.8% in Iceland.

Why was the GNE variant not identified through standard analyses of the WGS data, given this is a rare (ie MAF <2%) homozygous variant, in an established disease gene? Whilst not an exact phenotypic match, I'd expect to have been identified as a candidate?

Answer: We did indeed identify the GNE genotype through our WGS analysis, although we initially were unsure that this was the cause of this severe infant disease, based on the typically later onset and milder presentation of GNE-myopathy. In fact, it was the observation of a deficit of homozygous carriers at the population level that convinced us of the pathogenicity of this genotype. We thank the reviewer for highlighting the need to clarify this, and have now added the following paragraph to our chapter on GNE p.Asp378Tyr (page 8):

„Prior to acquiring this biological sample, we performed WGS for her non-consanguineous parents (meiotic distance = 16), searching for potential recessive genetic causes for her condition. We observed that the parents were both heterozygous carriers of the GNE p.Asp378Tyr variant, which is a known pathogenic variant in GNE-myopathy^{16,17}. GNE-myopathy, or Nonaka myopathy, is a progressive AR disease of muscle weakness with typical age of onset between 20 and 40 years (ClinVar Variation ID 283278)^{16,17}. The observed phenotype of the female infant did not fully align with the expected phenotype for GNE-myopathy, and it was unclear that it could be the cause of that lethal condition. However, WGS of a biological sample from her confirmed that she was homozygous for GNE p.Asp378Tyr, and the deficit of homozygous carriers in the population set supports the pathogenic role of this genotype.“

The CPSF3 variant appears to be on a haplotype that is unique to the Icelandic population. Have the authors considered estimating the age of this variant?

Answer: As the reviewer points out, the CPSF3 p.Gly468Glu variant is only detected in Iceland, i.e. is absent from gnomAD (N=141K). We note that the variant is also absent from a set of individuals of non-Icelandic ancestry who have also been WGS at deCODE (N=40K; mainly Danish, Swedish, Norwegian, Dutch, and US samples). The variant has reached a considerable frequency in the Icelandic population (340K inhabitants), carried by 1 in 126 Icelanders (1,261 heterozygous carriers out of 153,054 chip-genotyped and imputed Icelanders), indicating that it is not recent but likely originated many centuries ago in Iceland. The average pairwise meiotic distance between carriers is 14.78, compared to 15.28 in the general population, showing that the variant is widespread in the population and that tracing it back to a single ancestor is not feasible.

Reviewers' Comments:

Reviewer #2:

Remarks to the Author:

I thank the authors for their informative responses to my earlier comments. I am happy that any (very minor) concerns I had, have now been addressed.

Reviewers' comments:

Reviewer #2 (Remarks to the Author):

I thank the authors for their informative responses to my earlier comments. I am happy that any (very minor) concerns I had, have now been addressed.

Answer: We thank the reviewer for the positive feedback.